# Cu/CuO@ZnO Hollow Nanofiber Gas Sensor: Effect of Hollow Nanofiber Structure and P–N Junction on Operating Temperature and Sensitivity

**DOI:** 10.3390/s19143151

**Published:** 2019-07-17

**Authors:** Sung-Ho Hwang, Young Kwang Kim, Seong Hui Hong, Sang Kyoo Lim

**Affiliations:** Smart Textile Convergence Research Group, Daegu Gyeongbuk Institute of Science & Technology (DGIST), Daegu 42988, Korea

**Keywords:** electrospinning, photodeposition, ZnO hollow nanofiber, Cu, CuO, p–n junction, CO gas sensor

## Abstract

For the fast and easy detection of carbon monoxide (CO) gas, it was necessary to develop a CO gas sensor to operate in low temperatures. Herein, a novel Cu/CuO-decorated ZnO hollow nanofiber was prepared with the electrospinning, calcination, and photodeposition methods. In the presence of 100 ppm CO gas, the Cu/CuO-photodeposited ZnO hollow nanofiber (Cu/CuO@ZnO HNF) showed twice higher sensitivity than that of pure ZnO nanofiber at a relatively low working temperature of 300 °C. The hollow structure and p–n junction between Cu/CuO and ZnO would be considered to contribute to the enhancement of sensitivity to CO gas at 300 °C due to the improved specific surface area and efficient electron transfer.

## 1. Introduction

Carbon monoxide (CO), a dangerous byproduct of the incomplete combustion of fossil fuels, adversely affects human health because its high reactivity with blood erythrocytes results in oxygen deficiency in the body [1,2]. To monitor the toxic gas, a gas sensor is imperative due to concerns for industry safety requirements. With this in mind, metal oxide (SnO_2_, WO_3_, In_2_O_3_, ZnO, Fe_2_O_3_, and TiO_2_)-based gas sensors were intensively investigated for detecting hazardous gases (CO and H_2_S) and identifying their mechanism [3,4,5,6,7,8,9]. For example, the core mechanism of detecting H_2_S gas with a CuO/ZnO hybrid is destroying the p–n junction with the formation of metallic CuS in the presence of H_2_S gas [10]. On the other hand, in the case of CO gas, resistance changes by the oxidation of adsorbed CO through accepting the adsorbed oxygen ions at CuO/ZnO interfaces [11]. In this work, we focused on developing a CO gas sensor with good sensitivity, and ZnO was selected as the base material among the metal oxides because of its n-type nature and good thermal stability [12]. Another important issue is lowering the operating temperature of the sensor to improve durability and reduce the danger on account of high operating temperatures (~400 °C) [13,14]. For example, spike-shaped CuO/ZnO nanorods were developed for lowering the operating temperature of detecting 100 ppm of CO gas [15]. CuO-ZnO/Al_2_O_3_ ternary material that can operate at 175 °C was also developed to detect 200 ppm of CO gas [16].

In this study, two combined strategies, designing a hollow ZnO structure and controlling the depletion layer by metal oxide deposition, are suggested to reduce the operating temperature of the sensor. The hollow structure of ZnO has high potential for improving the gas response since hollow structures (inner-structure-controlled 1D nanostructure) would provide more active sites to adsorb CO gas, as well as gas-diffusion channels promoting faster electron transfer than other nanostructures [17,18]. The hollow structure can be prepared by three electrospinning strategies (coaxial, phase-separation, and fiber-template methods) [19,20,21]. The fiber-template method is more appropriate than the other strategies because fiber shell thickness and inner diameter are easily controlled by calcination. In addition, metal or metal oxide deposition can be applied straightforwardly on the rigid fiber template. It is well known that deposited metal nanoparticles help to adsorb oxygen molecules and then dissociate them on the surface of metal/metal oxide hybrid nanoparticles. These reactions improve gas sensitivity and charge transfer by reducing potential barriers at the grain boundaries. Thus, the depletion layer of the gas-sensing material, which can significantly affect gas sensitivity, was controlled by the formation of a p–n heterojunction through the photodeposition of a p-type metal oxide (CuO) on the surface of an n-type metal oxide (ZnO) [22]. With this in mind, in order to lower the operating temperature of the CO gas sensor with good sensitivity, new p–n heterojunction material was prepared. First, a polymeric solution containing Zn precursor was electrospun and annealed to obtain the ZnO hollow nanofiber. Second, Cu and CuO nanoparticles were photodeposited on the surface of the ZnO hollow nanofiber with different concentrations of a Cu precursor solution.

## 2. Materials and Methods

To design a CO gas sensor with a new structure, the ZnO hollow nanofiber was prepared with the electrospinning and calcination method, as previously reported [23,24]. Zinc acetate (ZnAc, Daejung, 99%, Korea) was dispersed in a 10 wt.% solution of polyacrylonitrile (PAN, Aldrich (St. Louis, MO, USA); molar weight, 150,000)/*N,N*-dimethylformamide (DMF, Aldrich, 99%) by stirring, and the mixture was sonicated to ensure good dispersion. The mass ratio of ZnAc to PAN was 1:1. A yellow viscous ZnAc/PAN gel was placed inside the syringe pump equipped with a 25-gauge stainless steel needle, and then was fed to the collector (distance between needle and collector was 15 cm) at a constant flow rate of 20 μL min^−1^ and electric field of 20 kV, which finally led to the webs of ZnAc/PAN nanofibers. To obtain ZnO hollow nanofibers, the webs of ZnAc/PAN nanofibers were placed in a box furnace and calcined in air at different temperatures (400, 500, 600, and 700 °C) for 4 h. Ramp rate was 5 °C min^−1^. As-prepared samples were denoted by ZnO NF-400, ZnO NF-500, ZnO NF-600 and ZnO NF-700, respectively.

To prepare a low-temperature-operated CO gas sensor, a combination of Cu and CuO particles were photodeposited on the ZnO hollow nanofiber using the Cu precursor, copper acetate (CuAc_2_, Aldrich, 99%) [25]. The amount of Cu and CuO on the ZnO hollow nanofibers was controlled by experimenting with different concentrations (0.1, 0.5, 1 mM) of CuAc_2_ in 10 vol.% of methanol (Daejung, 99%) aqueous solution. The as-prepared ZnO hollow-nanofiber (1 g L^−1^) samples were dispersed and stirred in CuAc_2_/methanol solutions for 30 min to allow the adsorption of Cu cations on the ZnO hollow-nanofiber surface. After that, the sample was exposed to an 8 W UVB lamp (λ = 365 nm) for 30 min to reduce Cu cations. Finally, different amounts of Cu- and CuO-photodeposited ZnO hollow nanofibers (Cu/CuO@ZnO HNF) were obtained. The samples prepared by using 0.1, 0.5, and 1 mM of CuAc_2_ are denoted as Cu/CuO(0.1 mM)@ZnO HNF, Cu/CuO(0.5 mM)@ZnO HNF, and Cu/CuO(1 mM)@ZnO HNF, respectively.

Morphological and elemental-mapping images of the as-prepared ZnO NFs and Cu/CuO@ZnO HNFs were obtained by using a scanning electron microscope (SEM, Hitachi, SU-8020, Tokyo, Japan) equipped with an energy-dispersive X-ray spectrometer. X-ray diffraction (XRD) patterns of the as-prepared samples were acquired with an X-ray diffractometer (PANalytical, Empyrean, 60 kV, Malvern, UK) using Cu–Kα1 radiation (λ = 1.5405 Å) and a quartz monochromator. A photoluminescence experiment was performed with a 325 nm He–Cd laser (Kimon, 1 K, 50 mW). The Brunauer–Emmett–Teller (BET) surface area of samples were calculated from nitrogen adsorption–desorption isotherms at 77 K (ASAP2020, Micrometrics). The effective surface areas were estimated at a relative pressure (P/P_0_) ranging from 0.06 to 1. The electrical conductivity of the samples was measured by using a 4-point probe system (AIT, CMT-SR2000N).

The sensor was fabricated by dropping ZnO NF or Cu/CuO@ZnO HNF gels on a sapphire substrate with prepatterned Pt electrodes. The ZnO NF or Cu/CuO@ZnO HNF gels were made by grinding 10 mg of ZnO NFs or Cu/CuO@ZnO HNFs with 0.25 mL of deionized water in an agate mortar. A precise autopipette (Eppendorf) was then used to deposit the solution. After dropping 10 μL of ZnO NFs or Cu/CuO@ZnO HNFs, the substrate was dried in air. The sensor element was heated at 400 °C for 1 h to prevent crystal growth at sensing temperatures (50–400 °C).

The properties of the ZnO NFs or Cu/CuO@ZnO HNFs must be well-analyzed to optimize their characteristics for CO gas-sensing applications. Sensing measurements were performed on a computer-based testing apparatus (PXI-DAQ system, National Instruments, Austin, TX, USA), and gas concentration was controlled by a precalibrated mass-flow meter (Brooks 5850E, Hatfield, UK). A source measure unit (4200SCS, Keithley, Beaverton, OR, USA) was used for acquisition of the electrical signal, while a power supply (E3631A, Agilent, Santa Clara, CA, USA) was employed to bias the sensor’s built-in back heater. The gas response is defined as the percentage change in resistance of the sensor upon CO exposure ((R_a_ − R_g_)/R_a_) × 100 (%), where R_a_ and R_g_ are resistances in air and in the presence of CO, respectively.

## 3. Results and Discussion

### 3.1. CO Gas Sensitivity of ZnO Hollow Nanofiber

#### 3.1.1. Effect of Annealing Temperature and Time on ZnO Nanofiber Morphology

In CO gas-sensing tasks, controlling sensor surface area and pore-size distribution is crucial. They can be controlled by changing the morphology of the gas-sensing materials, so it was necessary to investigate the influence of fabrication conditions (e.g., annealing temperature and time) on the morphology of ZnO nanofibers. With this in mind, pristine ZnAc/PAN nanofibers were calcined to obtain ZnO hollow nanofibers with different temperatures (400 ~ 700 °C) and time (1 ~ 4 h). Then, the inner morphologies of the obtained nanofibers were compared.

At first, the cross-sectional view of ZnO NF-400 seems to be fully filled without hollowness (Figure 1a). However, a hollow structure was observed in all other samples, ZnO NF-500, ZnO NF-600, and ZnO NF-700 (Figure 1b–d). As shown in Figure 2, when the ZnO NF was calcined at 500 °C for 1 h, a narrow channel (diameter of 30 nm) inside ZnO NF was formed, and further calcination (2 ~ 4 h) seemed to make a larger cavity (darkest area of image) inside ZnO NFs in proportion with calcination time. According to the results of Figure 1 and Figure 2, it was confirmed that hollow nanofiber structures started to form in calcination at 500 °C for 2 h or longer. To investigate the elemental composition and crystallinity of as-prepared ZnO NFs, the XRD spectra of the samples are presented in Figure 3a. The XRD patterns of as-prepared samples reveal that all the calcined nanofibers were composed of ZnO, and it was found that the characteristic peak intensity of ZnO increased by raising the calcination temperature due to higher crystallinity (Figure 3a). Therefore, it could be understood that calcination at 500 °C or more for 2 h or longer formed a hollow inside the pristine nanofibers throughout the thermal decomposition of the PAN component within the pristine ZnAc/PAN nanofibers. As shown in Figure 2, according to the increase of calcination temperature, the calcined inner diameters of the nanofibers were increased, and the nanofiber simultaneously came to form an outer shell with the higher density of ZnO. The tendency of the hollow structure of the ZnO nanofibers was highly related to phase separation (called the bleed-out phenomenon), which is a gradient rearrangement of metal oxides during the calcination of nanofibers [26].

#### 3.1.2. Effect of Calcination Temperature on ZnO Nanofiber Sensitivity

As described earlier, the CO gas sensitivity of ZnO nanofibers was evaluated by measuring resistance changes (R_a_ − R_g_/R_a_ × 100 (%)), where R_a_ is sample resistance in air and R_g_ is sample resistance in CO gas), of ZnO nanofibers in different operating temperatures (25 ~ 450 °C). CO sensing of ZnO resulted from surface chemical reactions. The surface chemical reaction between the chemisorbed oxygen species and the adsorbed CO gas molecules led to the resistance variation of the gas-sensor material. Generally, sensitivity of semiconductor metal oxide sensors is reported to improve in proportion with the operating temperature due to more electron density in the conduction band but, beyond the optimal operating temperature, desorption of the target gas would prevail on such electronic contribution, resulting in sensitivity reduction [3]. From this point of view, it was necessary to evaluate the sensitivity of our samples to CO gas, and to determine the best among as-prepared ZnO nanofibers and its optimal operating temperature. In Figure 3b, the responses of as-prepared ZnO nanofibers to CO gas depending on temperature are depicted. The resistance of all samples decreased in the presence of CO gas compared with those in air. Among the as-prepared ZnO nanofibers, ZnO NF-500 showed the best response (81.9% for 100 ppm CO gas) at an operating temperature of 400 °C (Figure 3b). To further investigate the correlation between the surface characteristics and CO gas sensitivity of the ZnO NF samples, photoluminescence spectra were measured, as shown in Figure 3c. In the photoluminescence spectra of the ZnO nanofiber, it was reported that the long-wavelength emission bands were attributed to defective emissions [27]. Among these emission bands, green emission bands (500–550 nm) are often attributed to singly ionized oxygen vacancies and neutral oxygen vacancies located on the surface, although this assignment is controversial. Yellow emission bands (550–600 nm) and orange-red emission bands (600–700 nm) are typically attributed to interstitial oxygen located on the subsurface. From the Figure 3c, it was shown that ZnO NF-500 exhibited the largest integrated area of PL intensity with long wavelength emission bands, from 500 to 800 nm, which showed that these were the most oxygen vacancies and interstitial oxygens in ZnO NF-500. From closer examination of the morphology and photoluminescence spectra of the samples, the hollow structure and presence of surface defects and interstitial oxygen of ZnO NF-500 could be expected to be more beneficial in reacting with CO gas than other samples due to more active sites. In addition, the oxygen species can influence operating temperature, which is highly related to the gas sensitivity of the ZnO nanofibers. The oxygen species existed differently depending on temperature. O_2_^−^ and O^−^ ions can exist at 100–300 °C [5]. However, O^2−^, resulting in a larger resistance change than O_2_^−^ or O^−^, can be chemisorbed on the inner and outer surface of ZnO NF-500 in >300 °C temperatures, which led to maximum gas sensitivity at 400 °C operating temperature. The ZnO NF-500 also showed excellent repeatability in CO gas response (Figure 3d). Therefore, we can assume that the optimal hollow structures of the ZnO nanofiber for CO gas sensing can be fabricated by calcination of a ZnAc/PAN nanofiber at 500 °C for 4 h.

### 3.2. CO Gas Sensitivity of Cu/CuO@ZnO HNF

#### 3.2.1. Effect of CuAc_2_ Concentration on ZnO Hollow-Nanofiber Morphology

As a second step to develop a CO gas sensor that can operate at lower temperatures, new p–n heterojunction materials (Cu/CuO@ZnO HNF) were prepared by photodepositing Cu or CuO on the surface of ZnO NF-500. To determine the best fabrication condition of a metal-semiconductor hybrid for CO gas sensing, three different concentrations of CuAc_2_ solutions (1, 0.5, 0.1 mM) were used. The morphology and elemental composition of the Cu/CuO@ZnO HNFs are shown in Figure 4 and Table 1. It was observed that Cu and CuO particles were deposited on the ZnO hollow nanofiber, showing different morphologies depending on the concentration of the Cu precursor solution. In Cu/CuO(0.1 mM)@ZnO HNF (Figure 4b), Cu and CuO with a seedlike shape started to form on the surface of the ZnO hollow nanofibers. In Cu/CuO(0.5 mM)@ZnO HNF (Figure 4c), Cu and CuO particles with a size of 50 nm were observed to be uniformly deposited on the ZnO hollow nanofibers. However, in Cu/CuO(1 mM)@ZnO HNF (Figure 4d), Cu and CuO particles seemed to largely be aggregated around the edge of the ZnO hollow nanofibers. The amounts of Cu in the samples increased with the increase of the concentration of Cu precursor solutions. The ratio of Cu/Zn, the amount of Cu in the sample divide by the amount of Zn in the sample, was increased by up to 0.41 for the Cu/CuO(1 mM)@ZnO HNF sample. This indicates that Cu and CuO hybrid nanoparticles were formed on the surface of ZnO HNF by the photodeposition process.

The components of the metal-semiconductor hybrids were also confirmed by XRD patterns (Figure 5). The XRD patterns of Cu and CuO in Cu/CuO@ZnO HNFs consisted of three peaks at 42.3°, 61.45°, and 73.48°, which are indexed to the reflections from the Cu (111) plane, CuO (−113) plane, and CuO (311) plane, respectively [28,29]. All Cu/CuO@ZnO HNFs exhibited the wurtzite structure (JCPDS NO. 36-1451) of ZnO [30]. EDX and XRD results demonstrated that Cu and CuO particles that formed on the surface of the ZnO hollow nanofiber were deposited in an amorphous phase with a seedlike shape in Cu/CuO(0.1 mM)@ZnO HNF, in well-crystallized phase in Cu/CuO(0.5 mM)@ZnO HNF, and in a highly aggregated phase in Cu/CuO(1 mM)@ZnO HNF. It is therefore thought that Cu/CuO(0.5 mM)@ZnO HNF would show the best sensitivity among them due to the best crystallinity and dispersion of Cu/CuO particles.

#### 3.2.2. CO Gas Response of Cu/CuO-Photodeposited ZnO Hollow Nanofibers (Cu/CuO@ZnO HNF)

Developing a gas sensor that operates in low temperatures is very important to reduce energy consumption and to ensure measuring safety. The gas-sensing mechanism and kinetics can be influenced by operating temperature. In this study, response of Cu/CuO@ZnO HNFs to 100 ppm CO were evaluated at different operating temperatures, as shown in Figure 6. All the Cu/CuO@ZnO HNFs exhibited a higher response to CO gas than the ZnO HNFs under 300 °C, especially the highest in Cu/CuO(0.5 mM)@ZnO HNF. Moreover, it was confirmed that the response of Cu/CuO(0.5 mM)@ZnO HNF was around twice higher than that of ZnO NF-500 at 300 °C, and as much as that of ZnO NF-500 at 400 °C. As compared to other ZnO-based sensors for CO gas mentioned in the literature, which is shown in Table 2, the Cu/CuO(0.5 mM)@ZnO HNF showed a good response at 300 °C. To discuss the effect of Cu on the CO gas-sensing mechanism, it was necessary to identify the phase of Cu metal in Cu/CuO(0.5 mM)@ZnO HNF during CO gas sensing. Figure 7 shows the XRD patterns of the Cu/CuO(0.5 mM)@ZnO HNF and the annealed ones. It was observed that the intensity of the Cu (111) peak was decreased but remained, and a CuO (022) peak was newly formed by the oxidation of the Cu metal in the XRD patterns of annealed ones. This indicated that the sample changed a phase by thermal oxidation during CO gas sensing in a high operating temperature. More importantly, the phase of the Cu metal could exist at the optimal operating temperature.

Some studies showed that a Cu site on the surface of ZnO allows to improve CO gas adsorption at low temperatures [33,35,36,37]. It was reported that when Cu/ZnO hybrids are exposed to CO molecules, CO molecules prefer to adsorb on the Cu metal to form bonds between CO and Cu. Then, adsorption leads to the improvement of CO reactivity [33]. In addition, the P–N junction of CuO@ZnO hybrids allows larger resistance change by changing the thickness of the electron-depletion layers, which leads to higher sensitivity for CO gas than that of bare ZnO [26]. Therefore, when Cu/CuO@ZnO HNFs are exposed to CO gas, CO gas would be captured on the surface of both Cu and CuO, and then electrons could be transferred to the ZnO HNF. Electrons recombined with the hole in the CuO to form a thinner electron-depletion layer and cause resistance change [38]. On the other hand, it is notable that the response of the metal-semiconductor hybrid nanofiber, Cu/CuO@ZnO HNFs, decreases dramatically over 300 °C, while that of ZnO hollow nanofiber starts to decrease over 400 °C, as previously observed, revealing that the optimal operating temperature of a Cu/CuO@ZnO HNFs sensor for CO gas was lowered to 300 °C compared with the 400 °C of ZnO NF. The decrease of sensitivity was attributed to the decrease of electron mobility of the Cu metal and the desorption acceleration of CO molecules at higher temperatures. The specific surface area was measured and is listed in Table 3. Generally, a high specific surface area of sensing material has positive effects on the sensing response. The specific surface area of Cu/CuO(0.5 mM)@ZnO HNF was 24.1 m^2^g^−1^, which is 1.9 times higher than that of ZnO NF-500 (12.4 m^2^ g^−1^). The hollow structure of Cu/CuO@ZnO HNF could also be responsible for the enhancement of gas-sensing performances. The hollow structure of the Cu/CuO@ZnO HNF provides a channel for the diffusion of gas molecules. Moreover, it provides more active sites and improves the kinetics of surface chemical reactions. Cu could contribute to the sensitivity and operating temperature of the CO gas sensor with a competition effect of intrinsic electrical conductivity and electrical mobility, depending on temperatures. Sensitivity could especially be improved by both the p–n junction and Cu with an efficient electron transfer between the components (CuO, ZnO HNF, and Cu) under 300 °C. However, with operating temperature raised higher, electron transfer could be hindered by electron scattering in Cu due to an increase of phonon concentration. In addition, desorption of CO gas on the surface of the sensor became frequent. Therefore, CO gas sensitivity was decreased above 300 °C. In conclusion, three factors, the p–n junction effect and the enhanced specific surface area due to the hollow structure, and electrical properties of the samples via Cu, would contribute to the improvement of CO sensitivity of Cu/CuO@ZnO HNFs at lower operating temperatures.

## 4. Conclusions

In summary, for a CO gas sensor in low operating temperatures, a novel Cu/CuO-photodeposited ZnO hollow nanofiber (Cu/CuO@ZnO HNF) was prepared and characterized. Then, its gas-sensing performance and that of a nonphotodeposited ZnO hollow nanofiber toward CO gas was studied. For 100 ppm of CO gas, the Cu/CuO(0.5 mM)@ZnO HNF showed twice higher sensitivity than that of a pure ZnO NF at 300 °C. The enhancement of its sensing performance could be attributed to the hollow structure and the p–n junction. The introduction of a hollow structure improves the specific surface area and active sites. The introduction of Cu/CuO brought efficient electronic transfer. These two factors resulted in significant enhancement in sensitivity, even in low operating temperatures.

## Figures and Tables

**Figure 1 sensors-19-03151-f001:**
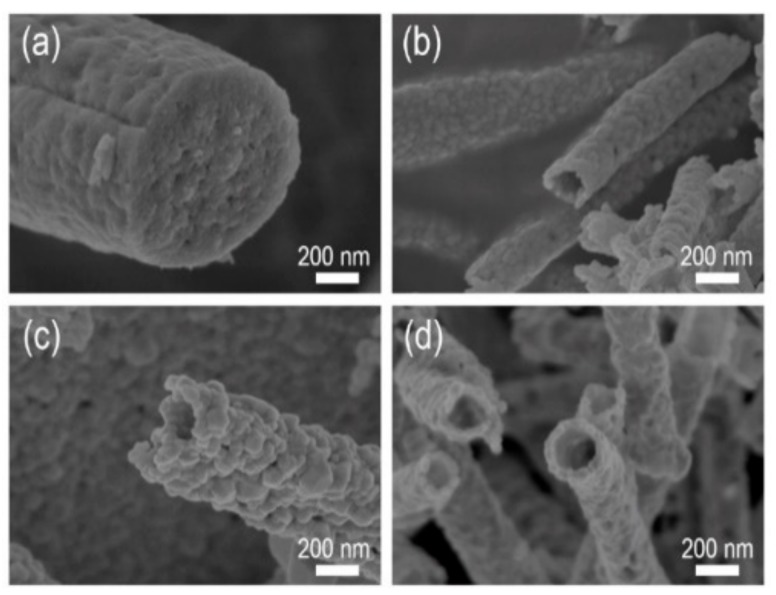
Scanning electron microscopy (SEM) images of ZnO nanofibers prepared at different annealing temperatures for 4 h: (**a**) 400 °C; (**b**) 500 °C; (**c**) 600 °C; (**d**) 700 °C.

**Figure 2 sensors-19-03151-f002:**
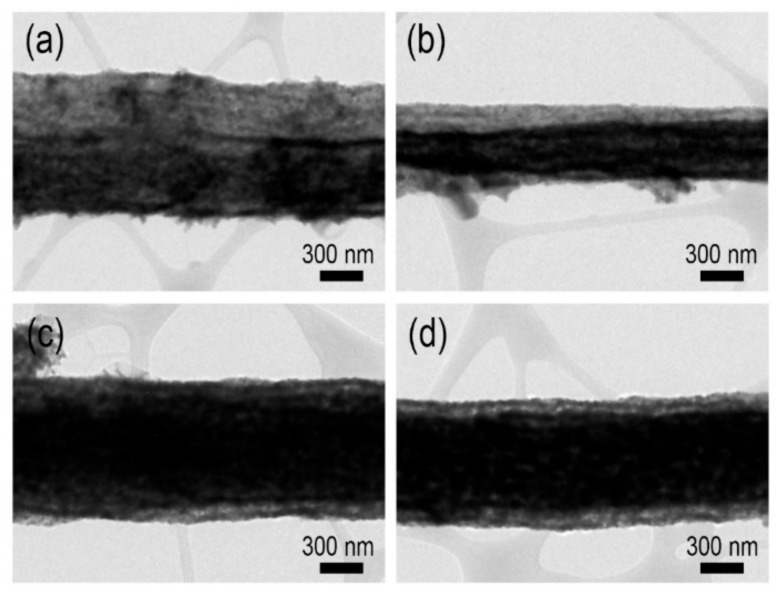
Transmission electron microscopy (TEM) images of ZnO NF-500 sample at different annealing times: (**a**) 1 h; (**b**) 2 h; (**c**) 3 h; (**d**) 4 h.

**Figure 3 sensors-19-03151-f003:**
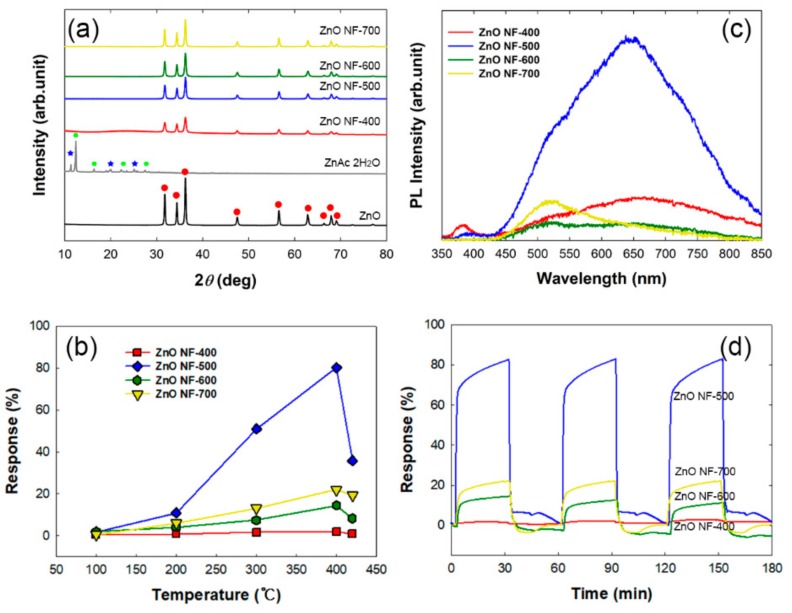
(**a**) X-ray diffraction (XRD) spectra of as-prepared ZnO nanofibers with different annealing temperatures for 4 h. Red circle, characteristic peaks of ZnO (JCPDS No. 36-1451); green hexagon and blue star, characteristic peaks of ZnAc (JCPDS No. 33-1464). (**b**) CO gas (100 ppm)-sensing responses depending on temperatures of as-prepared ZnO nanofibers with different annealing temperatures for 4 h. (**c**) Photoluminescence spectra of as-prepared ZnO nanofibers with different annealing temperatures for 4 h. (**d**) Repeatability test of the ZnO nanofibers as CO gas sensors.

**Figure 4 sensors-19-03151-f004:**
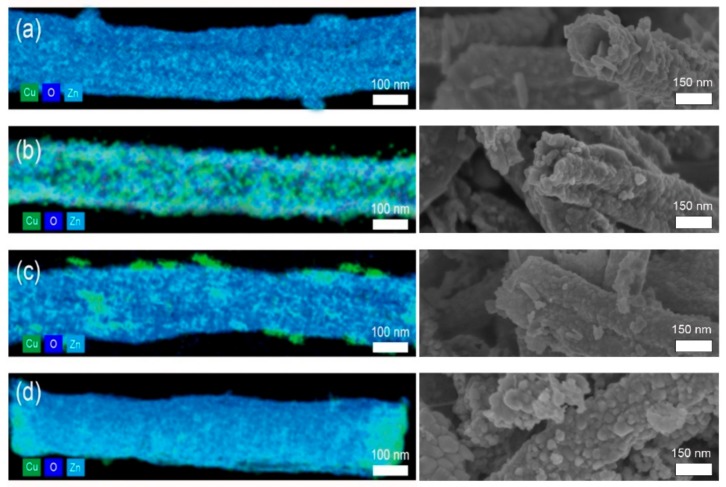
Energy dispersive X-ray spectroscopy (EDX) mapping images (left) and SEM images (right) of Cu/CuO-photodeposited ZnO hollow nanofibers (Cu/CuO@ZnO HNFs) with different concentrations of CuAc_2_: (**a**) 0 mM; (**b**) 0.1 mM; (**c**) 0.5 mM; (**d**) 1 mM.

**Figure 5 sensors-19-03151-f005:**
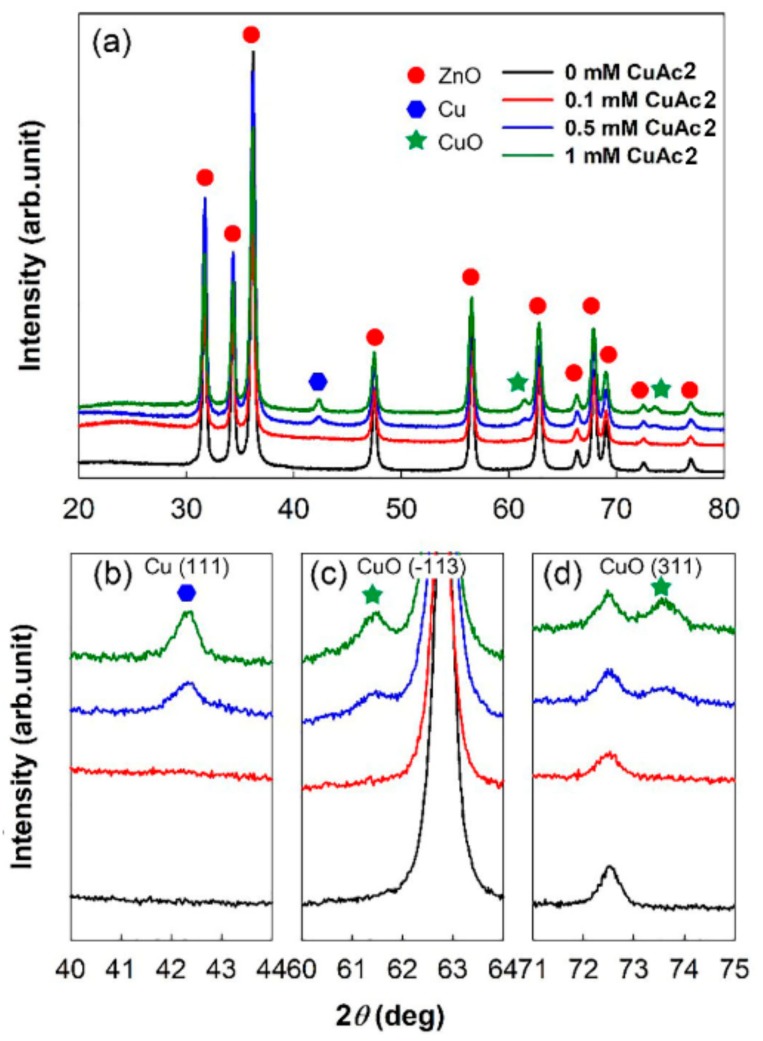
(**a**) XRD spectra of Cu/CuO@ZnO HNFs with different concentrations of CuAc_2_; (**b**) enlarged XRD spectra in a 2-theta range of 40°–44°; (**c**) enlarged XRD spectra in the 2-theta range of 60°–64°; (**d**) enlarged XRD spectra in the 2-theta range of 71°–75°.

**Figure 6 sensors-19-03151-f006:**
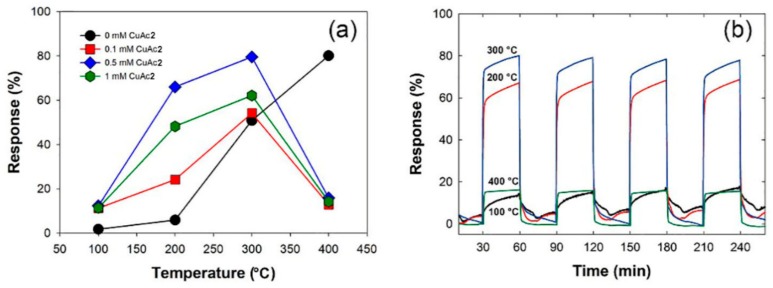
(**a**) CO gas-sensing (100 ppm) responses depending on temperatures of as-prepared Cu/CuO@ZnO HNFs; (**b**) repeatability test of Cu/CuO(0.5 mM)@ZnO HNF for CO gas in different working temperatures.

**Figure 7 sensors-19-03151-f007:**
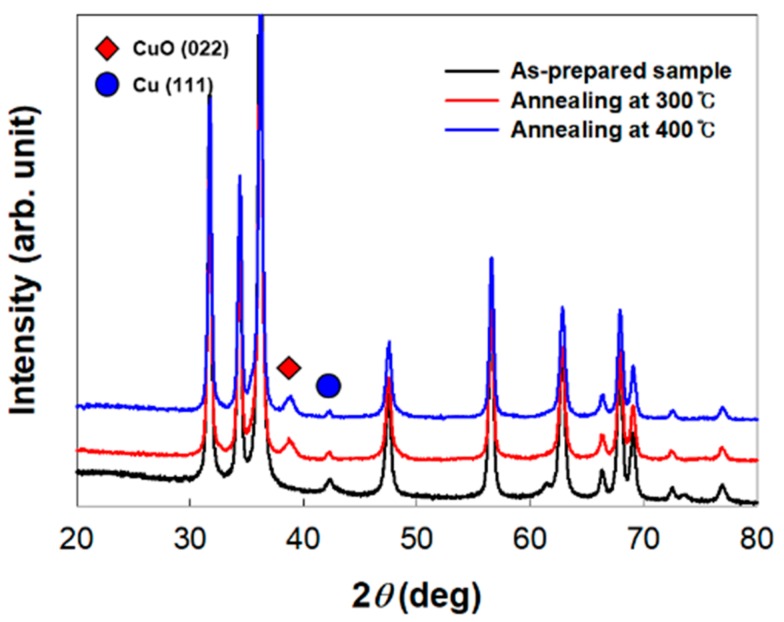
XRD patterns of as-prepared Cu/CuO(0.5 mM)@ZnO HNF and sample after annealing in air at different temperatures for 30 min. Red diamond, characteristic peak of CuO (022); blue circle, characteristic peak of Cu (111).

**Table 1 sensors-19-03151-t001:** Elemental composition of Cu/CuO@ZnO HNF.

Sample	Cu (%)	Zn (%)	Cu/Zn
ZnO NF-500	0	100	0
Cu/CuO(0.1 mM)@ZnO HNF	8.95	91.05	0.098 ± 0.007
Cu/CuO(0.5 mM)@ZnO HNF	19.85	80.15	0.248 ± 0.012
Cu/CuO(1 mM)@ZnO HNF	29.39	70.61	0.416 ± 0.005

**Table 2 sensors-19-03151-t002:** Comparison of sensing characteristics toward CO of some ZnO-based sensors.

Sample	CO (ppm)	Temperature (°C)	Response (%)	Reference
ZnO	200	300	90	[31]
Al/ZnO	200	300	50	[32]
CuO/ZnO	100	300	63	[15]
Cu/ZnO	100	300	69	[33]
ZnO-CuO/Al_2_O_3_	200	300	13	[16]
CuO/ZnO	250	175	11	[34]
Cu/CuO(0.5 mM)@ZnO HNF	100	300	78	This work
	100	200	65	

**Table 3 sensors-19-03151-t003:** Electrical conductivity and Brunauer–Emmett–Teller (BET) surface area of the samples.

Sample	Electrical Conductivity (S m^−1^)	BET Surface Area (m^2^ g^−1^)
ZnO NF-400 (no hollow)	-	7.8
ZnO NF-500	6.261 × 10^−6^	12.4
Cu/CuO(0.5 mM)@ZnO HNF	5.195 × 10^−5^	24.1

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
