# Peer review of "Cu/CuO@ZnO Hollow Nanofiber Gas Sensor: Effect of Hollow Nanofiber Structure and P–N Junction on Operating Temperature and Sensitivity"

_sensors, 2019, doi:10.3390/s19143151_

Round 1

Reviewer 1 Report

This work studies on the fabrication of Cu/Cuo/ZnO hollow nanofiber and its sensing performance toward CO gas. Several enhancement mechanisms were demonstrated in their work, including Cu induced electronic sensitization, hollow structure induced large surface area, and p-n junction induced signal amplification effect. The work is systematical and significant. Minor revision is recommended:

1. Besides the EDX mapping images, it is suggested to add SEM (or TEM) images of Cu/CuO@ZnO HNFs synthesized with different amount of CuAc2 in Figure 4, for better understanding.

2. The copper acetate should be marked as CuAc2 instead of CuAc.

3. The authors should check the caption of Figure 3. There was no TGA data.

4. The authors should polish the English writing to avoid some evitable mistake, e.g. there was repeated word “at” in 216th line.

Author Response

Responses to the Reviewers

Manuscript ID: sensors-528578

Title: Cu/CuO@ZnO Hollow Nanofiber Gas Sensor; Effect of Hollow Nanofiber Structure and P-N Junction on Operating Temperature and Sensitivity

Dear Reviewers

First of all, I’d like to express my sincere and thanks to Reviewers who read my manuscript with deep concerns and to valuable comments.

According to reviewer’s comments, manuscript was revised and checked to avoid some typical mistakes and expression or notation to cause misunderstanding.

We appreciate kind comments and suggestion of reviewer again. Our responses to these comments are listed below, and the manuscript have been accordingly revised. All the changes were represented in blue color in revised manuscript.

Thank you for your kindness and help

Sincerely yours

Response to Reviewer 1 comments

This work studies on the fabrication of Cu/Cuo/ZnO hollow nanofiber and its sensing performance toward CO gas. Several enhancement mechanisms were demonstrated in their work, including Cu induced electronic sensitization, hollow structure induced large surface area, and p-n junction induced signal amplification effect. The work is systematical and significant. Minor revision is recommended:

Comment 1.  Besides the EDX mapping images, it is suggested to add SEM (or TEM) images of Cu/CuO@ZnO HNFs synthesized with different amount of CuAc2 in Figure 4, for better understanding.

Response 1. Thank you for the valuable advice on our manuscript. As the reviewer pointed out, the SEM images of Cu/CuO@ZnO HNFs synthesized with different amount of CuAc2 were added in Figure 4, for better understanding. (Please see the attachment)

The revised figure captions as follow. 

Page 7, line 201 

Figure 4. EDX mapping images (left) and SEM images (right) of Cu/CuO photodeposited ZnO hollow nanofibers (Cu/CuO@ZnO HNFs) with different concentrations of CuAc2:(a) 0 mM; (b) 0.1 mM; (c) 0.5 mM; (d)1 mM.

Comment 2. The copper acetate should be marked as CuAc2 instead of CuAc.

Response 2. Thank you for your kind comment about our mistake. As the reviewer pointed out, the notation of copper acetate in the manuscript was revised as CuAc2 instead of CuAc.

Comment 3. The authors should check the caption of Figure 3. There was no TGA data.

Response 3. I am sorry for confusing you. When the manuscript was rewritten, the caption of Figure 3 was not corrected. As the reviewer pointed out, the caption of Figure 3 (a) was revised in accordance with the figures.

The revised figure caption is as follow.

Page 5, line 138

Figure 3. (a) XRD spectra of as-prepared ZnO nanofibers with different annealing temperatures for 4 h. Red circle; characteristic peaks of ZnO, (JCPDS No. 36-1451), green hexagon and blue star; characteristic peaks of ZnAC, (JCPDS No. 33-1464);  (b) CO gas (100 ppm) sensing responses depending on temperatures of as-prepared ZnO nanofibers with different annealing temperatures for 4 h; (c) Photoluminescence spectra of as-prepared ZnO nanofibers with different annealing temperatures for 4 h; (d) Repeatability test of the ZnO NF-500 as CO gas sensors.

Comment 4. The authors should polish the English writing to avoid some evitable mistake, e.g. there was repeated word “at” in 216th line.

Response 4. The manuscript was totally revised with better expressions containing example which the reviewer pointed out.

 (Please see the attachment)

Reviewer 2 Report

The article is devoted to the study of the sensor properties of Cu/CuO/ZnO nanocomposites in CO detection. I have a number of notes that concern both the motivation of the research, the characterization of materials and the interpretation of sensor properties.

 I do not understand how this sensor material can be interesting for detecting CO. These doubts are confirmed by low values of the sensor response when detecting a very high concentration of CO. Modification with copper oxide could lead to an increase in the response of ZnO toward H2S. The authors have to give a very serious justification for their research in the Introduction part.

In addition to the information about copper distribution over ZnO surface it is necessary to provide quantitative information on the total elemental composition of nanocomposites.

The article does not provide any data on the effect of copper on the electrical resistance of ZnO. All measurement results are presented only as a relative value of the sensor response. However, a comparison of the resistance values of pure and modified ZnO will allow us to conclude about the mechanism of the influence of copper / copper oxide on the sensor properties of ZnO.

I did not understand the idea of the mechanism of electronic sensitization. Usually, this mechanism is considered for palladium or silver oxides, which are reduced to metals upon interaction with CO that leads to a change in the barrier at the interface between the main semiconductor oxide and the modifier. What do the authors mean in this case (Cu/CuO/ZnO)?

Author Response

Responses to the Reviewers

Manuscript ID: sensors-528578

Title: Cu/CuO@ZnO Hollow Nanofiber Gas Sensor; Effect of Hollow Nanofiber Structure and P-N Junction on Operating Temperature and Sensitivity

Dear Reviewers

First of all, I’d like to express my sincere and thanks to Reviewers who read my manuscript with deep concerns and to valuable comments.

According to reviewer’s comments, manuscript was revised and checked to avoid some typical mistakes and expression or notation to cause misunderstanding.

We appreciate kind comments and suggestion of reviewer again. Our responses to these comments are listed below, and the manuscript have been accordingly revised. All the changes were represented in blue color in revised manuscript.

Thank you for your kindness and help

Sincerely yours

Response to Reviewer 2 comments

The article is devoted to the study of the sensor properties of Cu/CuO/ZnO nanocomposites in CO detection. I have a number of notes that concern both the motivation of the research, the characterization of materials and the interpretation of sensor properties.

Comment 1. I do not understand how this sensor material can be interesting for detecting CO. These doubts are confirmed by low values of the sensor response when detecting a very high concentration of CO. Modification with copper oxide could lead to an increase in the response of ZnO toward H2S. The authors have to give a very serious justification for their research in the Introduction part.

Response 1. Thanks for reviewer’s valuable comment. Generally, the concentration of CO gas at atmosphere is known to be about 20 ppm and some early symptoms such as headache begin to occur over 200 ppm. And so, at a large number of literatures, it was reported to detect CO gas at several hundred ppm. We evaluated the CO gas sensitivity with 100 ppm, of which the concentration was lower than that commonly used for evaluating the sensitivity. And therefore, compared with the CO gas response in other literatures, the measured CO sensitivity could be thought to be meaningful.

In addition, as the reviewers pointed out, modification with other metal or metal oxide is very important and meaningful in our manuscript. We add some examples to explain the application of ZnO for detecting hazardous gases (CO and H2S) and revised the expressions to clarify the purpose of this researches in the introduction part.

The related general introduction is revised in the manuscript as follow.

Page 1, line 25

~ With this in mind, metal oxides(SnO2, WO3, In2O3, ZnO, Fe2O3, and TiO2)-based gas sensors were intensively investigated for detecting hazardous gases (CO and H2S) and identifying their mechanism [3-9]. For example, the core mechanism of detecting H2S gas by CuO/ZnO hybrid is destroying the p-n junction by the formation of metallic CuS in presence of H2S gas [10]. On the other hands, in the case of CO gas, resistance changes by oxidation of adsorbed CO through accepting the adsorbed oxygen ions at the CuO/ZnO interfaces [11]. Especially, in this work, we focused to develop CO gas sensor with higher sensitivity and ZnO was selected to the base material among the metal oxides, because of n-type nature and good thermal stability [12]. Another issue, lowering the operating temperature of the sensor was also important to improve the durability and reduce the danger on account of high operating temperature (~ 400 °C) [13, 14]. For example, spike-shaped CuO/ZnO nanorods was developed for lowering working temperature to detect 100 ppm CO gas [15]. CuO-ZnO/Al2O3 ternary material was also developed to detect 200 ppm CO gas that can operate at 175 °C [16]. ~

The references were added to clarify the purpose of this research.

Page 10, line 308

9.         Han, C.; Li, X.; Shao, C.; Li, X.; Ma, J.; Zhang, X.; Liu, Y., Composition-controllable p-CuO/n-ZnO hollow nanofibers for high-performance H2S detection. Sens. Actuators, B 2019, 285, 495-503.

10.       Datta, N.; Ramgir, N.; Kaur, M.; Kailasa Ganapathi, S.; Debnath, A. K.; Aswal, D. K.; Gupta, S. K., Selective H2S sensing characteristics of hydrothermally grown ZnO-nanowires network tailored by ultrathin CuO layers. Sens. Actuators, B 2012, 166-167, 394-401.

11.       Yoon, D. H.; Yu, J. H.; Choi, G. M., CO gas sensing properties of ZnO–CuO composite. Sens. Actuators, B 1998, 46, (1), 15-23.

12.       Tuncel, D.; Ökte, A. N., ZnO@CuO derived from Cu-BTC for efficient UV-induced photocatalytic applications. Catal. Today 2019, 328, 149-156.

13.       Liu, Z.; Yang, T.; Dong, Y.; Wang, X., A room temperature VOCs gas sensor based on a layer by layer multi-walled carbon nanotubes/poly-ethylene glycol composite. Sensors 2018, 18, (9), 3113.

14.       Nandy, T.; Coutu, R. A.; Ababei, C., Carbon monoxide sensing technologies for next-generation cyber-physical systems. Sensors 2018, 18, (10), 3443.

15.       Rai, P.; Jeon, S.-H.; Lee, C.-H.; Lee, J.-H.; Yu, Y.-T., Functionalization of ZnO nanorods by CuO nanospikes for gas sensor applications. RSC Adv. 2014, 4, (45), 23604-23609.

16.       Youl Bae, H.; Man Choi, G., Electrical and reducing gas sensing properties of ZnO and ZnO–CuO thin films fabricated by spin coating method. Sens. Actuators, B 1999, 55, (1), 47-54.

Comment 2. In addition to the information about copper distribution over ZnO surface it is necessary to provide quantitative information on the total elemental composition of nanocomposites.

Response 2. According to reviewer’s suggestion, the result of EDS analyses was added in Table 1. Quantitative analysis of Cu/CuO@ZnO HNF showed that Cu content increased with increasing CuAc2 concentration. These results support the uniform formation of Cu and CuO hybrid nanoparticles on the surface of ZnO HNF by the photodeposition process using CuAc2 solution, which has been confirmed in the results of XRD and EDS analyses.

The related discussion is listed in the manuscript as follow.

Page 6, line 197

 ~ nanofibers. The amounts of Cu in the samples increases with increasing of the concentration of Cu precursor solutions. It indicates that Cu and CuO hybrid nanoparticles are formed on the surface of ZnO HNF by the photodeposition process. ~

For better understanding of the effect of photodeposition process on the surface of ZnO HNFs, the results of quantitative analysis of Cu/CuO@ZnO HNF by using energy dispersive X-ray spectroscopy was added in the manuscript.

Page 6, line 204

Table 1. Elemental composition of the Cu/CuO@ZnO HNFs   

Sample

Cu   (at%)

Zn   (at%)

O   (at%)

ZnO   NF-500

-

46.2

53.8

Cu/CuO(0.1   mM)@ZnO HNF

5.8

59.0

35.2

Cu/CuO(0.5   mM)@ZnO HNF

10.7

43.2

46.1

Cu/CuO(1   mM)@ZnO HNF

17.4

41.8

40.8

Comment 3. The article does not provide any data on the effect of copper on the electrical resistance of ZnO. All measurement results are presented only as a relative value of the sensor response. However, a comparison of the resistance values of pure and modified ZnO will allow us to conclude about the mechanism of the influence of copper / copper oxide on the sensor properties of ZnO.

Response 3. We really agree with reviewer’s comment on electrical resistance. And so, in order to understand the mechanism of the influence of copper/copper oxide on the sensor properties of ZnO, we measured the electrical resistance of the pure ZnO HNFs and modified one (Cu/CuO(0.5 mM)@ZnO HNF). It was found that the electrical conductivity, which is one of important factors that can make an influence on the CO gas sensing performances, was improved ca. 9 times higher in Cu/CuO(0.5 mM)@ZnO HNFs as much as in pure ZnO HNFs. The increase of the electrical conductivity is thought to be originated from the formation of Cu and CuO hybrid nanoparticles on the ZnO HNF on account of higher electrical conductivity and charge transfer effect of Cu and CuO. Therefore, the enhancement of electrical conductivity of Cu/CuO@ZnO HNFs is considered to be related with sufficient formation of well-crystallized Cu and CuO hybrid nanoparticles on ZnO HNFs, which also influences on the sensitivity and operating temperature of CO gas sensor. In conclusion, it was considered that the sensitivity and operating temperature of CO gas sensor was governed by combination effect among morphology and electrical properties. Especially, the sensitivity could be improved by both p-n junction and Cu metal with the efficient electron transfer among the components (CuO, ZnO HNF and Cu metal) under 300 °C. However, at higher operating temperature, electron transfer could be hindered by electron scattering in Cu metal due to increasing of phonon concentration. In addition, desorption of CO gas on the surface of the sensor became frequently. Therefore, CO gas sensitivity would be inferred to get to decrease above 300 °C.

The related experimental procedure is listed in the manuscript as follow.

Page 2, line 88

~ (P/P0) ranging from 0.06 to 1. The electrical conductivity of the samples was measured by using 4-point probe system (AIT, CMT-SR2000N). The sensor ~

The related discussion is listed in the manuscript as follow.

Page 9, line 233

~ The higher sensitivity of Cu/CuO@ZnO HNF could be attributed to the improved adsorption of CO gas, enhanced electrical conductivity of the samples (Table 2), and favorable ~

Page 9, line 248

~ with 400°C of ZnO NF. The decreasing of the sensitivity was attributed to the decreasing of electron mobility of Cu metal and acceleration of desorption of CO molecules at higher temperatures. On the other hand, ~

Page 9, line 256

Moreover, it also provides more active sites and improves the kinetics of chemical reactions on the surface. The Cu metal could contribute to the sensitivity and operating temperature of CO gas sensor with competition effect of intrinsic electrical conductivity and electrical mobility depending on the temperatures. Especially, the sensitivity could be improved by both p-n junction and Cu metal with the efficient electron transfer among the components (CuO, ZnO HNF, and Cu metal) under 300 °C. However, with operating temperature raised higher, electron transfer could be hindered by electron scattering in Cu metal due to increasing of phonon concentration. In addition, desorption of CO gas on the surface of the sensor became frequently. Therefore, CO gas sensitivity was decreased above 300 °C. In conclusion, three factors, the p-n junction effect and the enhanced specific surface area due to the hollow structure, and electrical properties of the samples via Cu metal, would contribute to the improvement in CO sensitivity of Cu/CuO@ZnO HNFs.

The results of comparison of electrical conductivity among the Cu/CuO@ZnO HNFs samples was added in the manuscript.

Page 9, line 267

Table 2. Electrical conductivity and BET surface area of the samples.

Sample

Electrical conductivity

(S m-1)

BET surface area

(m2 g-1)

ZnO NF-400 (no hollow)

-

7.8

ZnO NF-500

6.261 × 10−6

12.4

Cu/CuO(0.5 mM)@ZnO HNF

5.195 × 10−5

24.1

Comment 4. I did not understand the idea of the mechanism of electronic sensitization. Usually, this mechanism is considered for palladium or silver oxides, which are reduced to metals upon interaction with CO that leads to a change in the barrier at the interface between the main semiconductor oxide and the modifier. What do the authors mean in this case (Cu/CuO/ZnO)?

Response 4. The mechanism of CO gas sensing in Cu/CuO@ZnO HNFs might be explained by combination effects via CuO@ZnO HNF and Cu metal. Firstly, CO molecules get adsorbed on the CuO surface and the oxygen ions on the ZnO surface and hence at the CuO@ZnO HNF interface, the adsorbed CO is oxidized into CO2 by accepting the adsorbed oxygen ions. Secondly, Cu metal in Cu/CuO@ZnO HNFs accelerates adsorption of CO molecules due to the coexistence of ZnO with Cu that can enhance the capability of Cu to adsorb CO molecules. The Cu site played a role to adsorb CO molecules at both low and high temperatures. When CO molecules are adsorbed on the Cu/CuO@ZnO HNFs, they are preferably adsorbed on the Cu sites to form bonds between them. The Cu–CO bonding consists of the donation of CO 5σ electrons to the metal and the back donation of π-electrons from d-orbitals of Cu to CO, and the adsorption results in the enhancement of the CO reactivity. Therefore, the CO adsorption took place both at the Cu sites and CuO, and then CO molecules migrated from the Cu and CuO to the ZnO HNFs. In this way, the Cu and CuO sites enhanced the CO adsorption and thus the reaction of CO with oxygen species.

The related discussion is listed in the manuscript as follow.

Page 9, line 236

~ at low temperature [31-34]. It was reported that when the Cu/ZnO hybrids are exposed to the CO molecules, CO molecules preferred to adsorb on the Cu metal to form bonds between CO and Cu. Then, the adsorption leads to the improvement of CO reactivity [33]. In addition, the P-N junction of CuO@ZnO hybrids allow larger resistance change by changing the thickness of electron depletion layers, which would lead to higher sensitivity for CO gas than that of bare ZnO [26]. Therefore, when the Cu/CuO@ZnO HNFs are exposed to CO gas, CO gas would be captured on the surface of both Cu and CuO and then electrons could be transferred to ZnO HNF. The electrons recombined with the hole in the CuO to form thinner electron depletion layer and cause resistance change [35].

(Please see the attachment.)

Round 2

Reviewer 2 Report

Unfortunately, the authors could not significantly improve the article.

1. Response to the comment about the CO concentration and the sensor response values l is unconvincing. Recommended CO concentration levels can be found here 

WHO. Air Quality Guidelines for Europe, 2nd ed.; WHO: Geneva, Switzerland, 2000; Available online: http://www.euro.who.int/__data/assets/pdf_file/0020/123059/AQG2ndEd_5_5carbonmonoxide.PDF?ua=1 

If the authors want to prove that they have received a highly sensitive sensor material, it is necessary to give literary data for comparison. I am sure that the materials with higher sensitivity to CO are described. In my opinion, the authors should abandon this thesis and offer another motivation to study the CuO/ZnO system as a material for CO detection.

2. Lines 49-50. Bad wording. The deposition of metal on an n-type semiconductor cannot lead to the formation of a p-n junction.

3. Table 1. The EDS method does not allow the quantification of oxygen (especially up to a tenth of a percent). This column should be removed from the table. The results of the composition analysis can be presented as [Cu]/[Zn] ratio with indication of measurement error.

4. The most important note. The phase composition is presented for as prepared Cu/CuO/ZnO composite samples. During the preparation of sensor elements and sensor measurements the thermal heating to 400 C in the air (that is, in the presence of oxygen) has been used. This, for sure, led to the complete oxidation of copper nanoparticles to CuO. Then all the arguments of the authors about the influence of copper on the sensor properties of the composites lose their meaning. It is necessary to anneal the composites under these conditions and re-examinate them by XRD. If the phase of metallic copper is absent in the samples after annealing , the sensitization effect may be due to the reduction of CuO to Cu under the action of CO (this reaction is possible at 250-400 C).  This is an electronic sensitization effect similar to that described for n-type semiconductor oxides modified by Ag2O and PdO.

Author Response

Responses to the Reviewers

Manuscript ID: sensors-528578

Title: Cu/CuO@ZnO Hollow Nanofiber Gas Sensor; Effect of Hollow Nanofiber Structure and P-N Junction on Operating Temperature and Sensitivity

Dear Reviewers

First of all, I’d like to express my sincere and thanks to Reviewers who read my manuscript with deep concerns and to valuable comments.

According to reviewer’s comments, the manuscript was revised.

We appreciate kind comments and suggestion of reviewer again. Our responses to these comments are listed below, and the manuscript has been accordingly revised. The manuscript was revised with better expressions containing example which the reviewer pointed out. All the changes were represented in red color in the revised manuscript.

Thank you for your kindness and help

Sincerely yours

Response to Reviewer 2 comments_2

Unfortunately, the authors could not significantly improve the article.

Comment 1. Response to the comment about the CO concentration and the sensor response values l is unconvincing. Recommended CO concentration levels can be found here 

WHO. Air Quality Guidelines for Europe, 2nd ed.; WHO: Geneva, Switzerland, 2000; Available online: http://www.euro.who.int/__data/assets/pdf_file/0020/123059/AQG2ndEd_5_5carbonmonoxide.PDF?ua=1 

If the authors want to prove that they have received a highly sensitive sensor material, it is necessary to give literary data for comparison. I am sure that the materials with higher sensitivity to CO are described. In my opinion, the authors should abandon this thesis and offer another motivation to study the CuO/ZnO system as a material for CO detection.

Response 1. Thanks for the reviewer’s valuable comment and also for giving very detailed information about CO concentration level. WHO Air Quality Guidelines for Europe says that ‘global background concentrations of carbon monoxide range between 0.06 mg/m3 and 0.14 mg/m3 (0.05–0.12 ppm), and, in urban traffic environments of large European cities, the 8-hour average carbon monoxide concentrations are generally lower than 20 mg/m3 (17 ppm) with short-lasting peaks below 60 mg/m3 (53 ppm). And also, carbon monoxide concentrations can quite a little change according to various microenvironment condition such as ventilation, gas appliances, vehicles and etc. And so, in underground and multistory car parks, road tunnels, and enclosed ice arenas and various other indoor microenvironments, the mean levels of carbon monoxide can rise above 115 mg/m3 (100 ppm) for several hours, with short-lasting peak values that can be much higher, and, in homes with gas appliances, peak carbon monoxide concentrations of up to 60–115 mg/m3 (53–100 ppm) have been measured’. And so, I think that ‘atmosphere’ in our last response about CO concentration (‘Generally, the concentration of CO gas at atmosphere is known to be about 20 ppm’) could make reader confused. it would be more appropriate to use ‘ambient’ rather than ‘atmosphere’. Please, I hope you to understand our mistake.

Of course, in spite of the average CO concentration in these cities, it will be important to prepare a sensor with higher sensitivity since it will have to improve to a lower CO concentration level at cities. And also, in a healthy subject, the guideline said that, for nonsmoking, middle-aged and elderly population groups with documented or latent coronary artery disease from acute ischemic heart attacks, and the fetuses nonsmoking pregnant women, a COHb level of 2.5% should not be exceeded. Therefore, as the reviewer pointed out, I agreed with the necessity for the higher sensitive sensor.

According to the reviewer’s comment, we rewrote the manuscript in the viewpoint of a useful sensor at lower operating temperature rather than the materials with higher sensitivity to CO. In addition, the table was added to compare the sensing characteristics such as sensitivity and operating temperature toward CO gas with those reported in the literature.

The manuscript is revised as follow.

Page 1, line 30

Especially, in this work, we focused to develop CO gas sensor with good sensitivity and ZnO was selected to the base material among the metal oxides, because of n-type nature and good thermal stability [12]. ~

Page 2, line 52

With this in mind, in order to lower the operating temperature of the CO gas sensor with good sensitivity, new p-n heterojunction material was prepared.

Page 6, line 186

As a second step to develop the CO gas sensor which can operate at a lower temperature, new p-n heterojunction materials (Cu/CuO@ZnO HNF) were prepared by photodepositing Cu or CuO on the surface of ZnO NF-500.

Page 8, line 233

 ~ at 400 °C. As compared to other ZnO-based sensors for CO gas mentioned in the literature, which is shown in Table 2, the Cu/CuO(0.5 mM)@ZnO HNF showed a good response at 300 °C. ~

Page 10, line 286

In summary, for a CO gas sensor in low operating temperature, a novel Cu/CuO-photodeposited ZnO hollow nanofiber (Cu/CuO@ZnO HNF) has been prepared and characterized. Then, ~

Table 2 and references were added in the manuscript to compare the CO sensitivity of the sensor and that of previously reported ones.

Page 9, line 243

Table 2. Comparison of the sensing characteristics toward CO of some ZnO-based sensors.

Sample

CO (ppm)

Temperature (°C)

Response (%)

Reference

ZnO

200

300

90

[31]

Al/ZnO

200

300

50

[32]

CuO/ZnO

100

300

63

[15]

Cu/ZnO

100

300

69

[33]

ZnO-CuO/Al2O3

200

300

13

[16]

CuO/ZnO

250

175

11

[34]

Cu/CuO(0.5 mM)@ZnO HNF

100

300

78

This work

100

200

65

Page 13, line 378

31.       Krishnakumar, T.; Jayaprakash, R.; Pinna, N.; Donato, N.; Bonavita, A.; Micali, G.; Neri, G., CO gas sensing of ZnO nanostructures synthesized by an assisted microwave wet chemical route. Sens. Actuators, B 2009, 143, (1), 198-204.

32.       Chang, J. F.; Kuo, H. H.; Leu, I. C.; Hon, M. H., The effects of thickness and operation temperature on ZnO:Al thin film CO gas sensor. Sens. Actuators, B 2002, 84, (2), 258-264.

33.       Gong, H.; Hu, J. Q.; Wang, J. H.; Ong, C. H.; Zhu, F. R., Nano-crystalline Cu-doped ZnO thin film gas sensor for CO. Sens. Actuators, B 2006, 115, (1), 247-251.

34.       Ghosh, A.; Bhowmick, T.; Labhasetwar, N.; Majumder, S. B., Catalytic oxidation and selective sensing of carbon monoxide for sense and shoot device using ZnO–CuO hybrids. Materialia 2019, 5, 100177.

Comment 2. Lines 49-50. Bad wording. The deposition of metal on an n-type semiconductor cannot lead to the formation of a p-n junction.

Response 2. The manuscript was revised with better expressions containing example which the reviewer pointed out.

Page 1, line 37

In this study, two combined strategies, designing a hollow structure of ZnO and controlling the depletion layer by metal oxide deposition, ~

Page 2, line 50

Thus, the depletion layer of the gas sensing material, which can significantly affect the gas sensitivity, was controlled by the formation of p-n heterojunction through photodeposition of p-type metal oxide (CuO) on the surface of n-type metal oxide (ZnO) [22]. 

Comment 3. Table 1. The EDS method does not allow the quantification of oxygen (especially up to a tenth of a percent). This column should be removed from the table. The results of the composition analysis can be presented as [Cu]/[Zn] ratio with indication of measurement error.

Response 3. According to the reviewer’s suggestion, the result of EDS analyses was revised in Table 1. Quantitative analysis of Cu/CuO@ZnO HNF showed that Cu content increased with increasing CuAc2 concentration. These results support the uniform formation of Cu and CuO hybrid nanoparticles on the surface of ZnO HNF by the photodeposition process using CuAc2 solution, which has been confirmed in the results of XRD and EDS analyses.

The related discussion is revised in the manuscript as follow.

Page 6, line 197

 ~ nanofibers. The amounts of Cu in the samples increases with increasing the concentration of Cu precursor solutions. The ratio of [Cu]/[Zn] is increased up to 0.41 for Cu/CuO(1 mM)@ZnO HNF sample. It indicates that Cu and CuO hybrid nanoparticles are formed on the surface of ZnO HNF by the photodeposition process. ~

For a better understanding of the effect of the photodeposition process on the surface of ZnO HNFs, the results of quantitative analysis of Cu/CuO@ZnO HNF by using energy dispersive X-ray spectroscopy was revised in the manuscript.

Page 7, line 205

Table 1. Elemental composition of the Cu/CuO@ZnO HNFs   

Sample

Cu (at%)

Zn (at%)

[Cu]/[Zn]

ZnO NF-500

0

100

0

Cu/CuO(0.1 mM)@ZnO HNF

8.95

91.05

0.098 ±   0.007

Cu/CuO(0.5 mM)@ZnO HNF

19.85

80.15

0.248 ±   0.012

Cu/CuO(1 mM)@ZnO HNF

29.39

70.61

0.416 ±   0.005

Comment 4. The most important note. The phase composition is presented for as prepared Cu/CuO/ZnO composite samples. During the preparation of sensor elements and sensor measurements the thermal heating to 400 C in the air (that is, in the presence of oxygen) has been used. This, for sure, led to the complete oxidation of copper nanoparticles to CuO. Then all the arguments of the authors about the influence of copper on the sensor properties of the composites lose their meaning. It is necessary to anneal the composites under these conditions and re-examinate them by XRD. If the phase of metallic copper is absent in the samples after annealing, the sensitization effect may be due to the reduction of CuO to Cu under the action of CO (this reaction is possible at 250-400 C). This is an electronic sensitization effect similar to that described for n-type semiconductor oxides modified by Ag2O and PdO.

Response 4. Thank you very much for a good idea to check the effect of Cu and CuO. As the reviewer pointed out, to identify the phase transition of as-prepared Cu/CuO@ZnO HNF after CO gas sensing measurement, the XRD patterns of as-prepared Cu/CuO(0.5 mM)@ZnO HNF and that of annealed ones were measured. For the measurement, the sample was annealed in air at 300 °C and 400 °C for 30 min, respectively. In high operating temperature, the sample changed a phase by the thermal oxidation. It was observed that the intensity of Cu (111) peak was decreased but remained, and CuO (022) peak was newly formed by the oxidation of Cu metal in XRD patterns of annealed ones. It indicated that the phase of Cu metal might obviously exist even at optimal operating temperature. Therefore, we think that, as the reviewer pointed out, the electron transfer at p-n junction between CuO and ZnO due to the reduction of CuO to Cu under the action of CO would play an important major role and also the Cu metal might play a role in Cu/CuO@ZnO HNF for CO gas sensing by efficient CO gas adsorption and electron transfer.

The related discussion is revised in the manuscript as follow.

Page 1, line 16

The hollow structure and the p-n junction between Cu/CuO and ZnO would be considered to contribute to the enhancement of sensitivity to CO gas at 300°C, due to the improved specific surface area and efficient electron transfer.

Page 9, line 235

To discuss the effect of Cu on the CO gas sensing mechanism, it is necessary to identify the phase of Cu metal in Cu/CuO(0.5 mM)@ZnO HNF during the CO gas sensing. Figure 7 shows the XRD patterns of the Cu/CuO(0.5 mM)@ZnO HNF and annealed ones. It was observed that the intensity of Cu (111) peak was decreased but remained and CuO (022) peak was newly formed by the oxidation of Cu metal in XRD patterns of annealed ones. It indicated that the sample changed a phase by the thermal oxidation during the CO gas sensing in high operating temperature. More importantly, the phase of Cu metal might exist at the optimal operating temperature. Some ~

Page 11 line 292

The introduction of Cu/CuO brings efficient electronic transfer. These ~

Figure 7 is added in the manuscript.

Page 9, line 245

Figure 7. XRD patterns of as-prepared Cu/CuO(0.5 mM)@ZnO HNF and the sample after annealing in air at different temperatures for 30 min. Red diamond; the characteristic peak of CuO (022), Blue circle; the characteristic peak of Cu (111).

Round 3

Reviewer 2 Report

The article can be accepted in present form.